# Family Environment and Rural Child Development in Shanxi, China

**He Li * and Ziyang Liu**

Department of Global Business, Kyonggi University, Suwon 16227, Korea
* Correspondence: lihe@hrbipe.edu.cn

**Abstract:** The family environment in rural Northwest China has undergone significant changes as a result of the accelerated migration of rural populations. We constructed an indicator system for measuring the development of rural children in northwest China and use Shanxi as an example to conduct field research on the effects of family structure on children's education. We constructed a children's comprehensive development index for regression analysis using the principal component method. We also built a mediating effect model based on the importance of parents' involvement in their children's development. Then, we examined the influence mechanism between family structure and children's development in northwest rural areas. Finally, we found that family structure significantly affects the development of rural children in Shanxi Province, especially two-parent families. Furthermore, parental emotional involvement has a significant mediating effect on rural children's development. Due to low parental emotional involvement, children from absent families are less developed than children from intact families. Moreover, parental behavior plays a significant role in mediating the relationship between family structure and child development. Due to a lower level of parental behavioral involvement, children from absent families are less likely to achieve comprehensive development than children from intact families. Consequently, we should set up a comprehensive management system that integrates family, schools, and local communities for the sustainable development of children.

**Keywords:** family environment; child development; emotional participation; behavioral participation

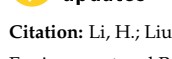

## 1. Introduction

Bernadette Daelmans et al. identified that the evidence across multiple disciplines is clear and compelling that investing in young children is the foundation of human and sustainable development: without the best start in life for all children, there is no foundation for sustainable societies [1]. Children are the future and hope of mankind. Children's development, including their sustainable education, has a multiplier effect in many global sustainable goals, which is related to sustainable economic and social development and the future of mankind [2]. Meanwhile, according to the 7th census, 64.1% of permanent residents aged 0–14 in Shanxi Province live in urban areas, and 35.9% live in rural areas. The household registration system in China leads to an unbalanced distribution of rural and urban economy and resources, which hinders the development of rural children, resulting in a wide gap between urban and rural children. In terms of both education and children in general, this development is not sustainable. As such, eliminating imbalances in child development has become an important area of research.

Due to the dual influence of the family planning policy and the traditional concept of family separation, there is a trend toward miniaturization and centralization in the rural family structure in China, and the large family that was once common in rural areas is rapidly fading away. As the majority of Shanxi is economically underdeveloped, migrating for work has become one of the few options for them to improve their quality of life. These changes have resulted in significant changes to the rural family structure in Shanxi. In



addition, the changes in the rural population's conception on marriage and love and the way of marriage, which caused by the migration of population and the "migrant economy", also have a great impact on the existing family structure within rural areas. The primary manifestation is the rising divorce rate in rural areas, and the increase of single-parent families and reorganized families. Thus, the national policy, the traditional idea, migration movements, and other factors have disrupted the original nuclear family structure of dual-parent upbringing in rural areas of Shanxi; various family structures have developed, including single-parent parenting and grandparenting that have a significant impact on the growth and development of rural children. There is evidence that the absence of a parent, the most important member of a family, may negatively affect the growth and development of a child. Those who experienced family breakdowns as children or who grew up in single-parent households perform worse in school and are more likely to be at a disadvantage in social development later in life [3]. Thus, family structure has become an important factor for solving the problem of unbalanced child development in Shanxi.

Several scholars have studied family structure, economic income level, family education methods, family parenting types, family atmosphere, family stability, and parents' expectations to understand their impacts on rural children's physical and mental health [4]. However, in the vast rural areas of our country, a large number of left-behind children have appeared as a result of the migration of populations between rural and urban areas. Therefore, domestic scholars' perspectives on the relationship between family and rural children's development mainly focus on the impact of parents' leaving on these children.

Migrant workers have affected the traditional "iron triangle" of rural family structure, which is detrimental to the healthy development of rural children. Additionally, left-behind children have a worse nutritional health status than non-left-behind children in rural areas. Researchers have found that children left behind by working parents are more likely to become ill or suffer from chronic diseases [5–7].

According to the perspective of rural children's educational achievements, given the large gap between urban and rural educational resources, rural children's educational development is in a relatively disadvantageous position, and the increasingly complicated rural family structure also contributes to their educational achievement. It has been shown in some studies that large-scale labor transfers have adversely affected rural family structures, resulting in poor academic performance for children [8,9]. Additionally, scholars have demonstrated that different lengths of parents' out-of-home time and different parents going out have different effects on the academic performance of their children [10,11].

In terms of children's behavioral development, current academic research findings can roughly be divided into two categories: limited impact and unlimited impact. It is clear from both points that there is a link between family structure and the development of child behavior. Based on the view of infinite influence, a family structure changes due to factors such as parents leaving or divorcing, which causes family functions to become difficult to exert through good parent-child interaction [12,13]. In the view of limited impact, the changing family structure due to parents leaving and divorce does have a negative impact on children's behavior, but the effects of school, society, and the children's self-control are equally inexorable [14,15].

In terms of the social development of rural children, the family is the main source of growth and development, and the family structure has a certain influence on the social development of these children. Changes in family structure, such as the left-behind children, parental divorce, and remarriage can result in deviations in children's social cognition, which in turn can affect their future interpersonal skills [16]. Consequently, due to the absence of both parents and migrant workers in the rural areas of Shanxi China, the social development of local rural children is hindered.

In general, most rural child development studies focus on left-behind children, and little research has been conducted on the development of other types of rural children. It is even more rare to conduct research on the development of rural children in areas with frequent mobility such as the Northwest. In addition, changes in the traditional family

structure in China are not only influenced by the changing views of the contemporary youth population towards marriage and family, but also by many other non-marriage factors. Moreover, non-marriage factors include: structural factors such as the dual structure of urban and rural areas; institutional factors such as the traditional household registration system and the enrollment system associated with household registration; and cultural factors such as intergenerational support between parents and offspring.

In this paper, based on the development of rural children in Shanxi, we construct a micro-evaluation index system that can be used to comprehensively quantify the development of children and analyze the impact of the changing rural family structure in Shanxi on the children's development, thereby verifying the specific mechanisms through which family structure affects the development of rural children in Shanxi China. Further, policy recommendations are provided for enhancing the development of rural children.

## 2. Methods

In this section, the theoretical method and the method for constructing the rural child development evaluation indicator system framework are presented.

### 2.1. Theoretical Method

On one hand, in socialization theory based on structural functionalism, the family is considered a structure, and each member of the family plays a specific role. In contrast with non-biparental families, two-parent families are more likely to provide economic support, life care, behavioral supervision, and role models for children's growth and development. Thus, it is likely that the absence of either parent will have an impact on the child's socialization and other aspects of development. Some researchers argue that there is a greater likelihood that families with a lack of intimacy, chaotic roles, or no stable rules will have members who run away from home, suffer from physical or mental illnesses, and have children who misbehave [16]. Therefore, based on the socialization theory of structural functionalism, this paper argues that in rural areas in northwest China, changes in family structure from migrant work or parental divorce will cause parents to lose the opportunity to participate in all aspects of their children's development. The loss of complete family functions such as role demonstration and behavior supervision will cause children who grow up in a family with one absent parent or both absent parents to differ in their development status in health, education, and other areas.

On the other hand, according to Becker (1993) [17], household activities include both consumption and production activities. Family resource theory emphasizes the importance of human capital investment in the development of children. A family's time resources have a greater impact on a child's education, skills, health and other human capital accumulation. It must be noted that time is a valuable resource for families, and that activities such as raising children and increasing family income will occupy the limited available family time. Consequently, parents can devote different amounts of time to raising their children according to the structure of their family. Families with incomplete structures will be at a disadvantage when it comes to time investment. According to the family resource theory, parents who work in rural areas of northwest China will lose the opportunity to participate in their children's development and growth, resulting in a decrease in parental time investment in their children, as well as poor child development.

So, this paper will analyze the theoretical framework that family structure affects children's development in combination with the theory of family resources and socialization and will set up an evaluation index system for rural children's development by choosing reasonable indicators. Then, based on the collected survey data and the established indicator system, this paper evaluates the current development of rural children in Northwest China. Finally, this paper proposes an econometric model for empirically studying how family structure affects the development of children in rural Northwest China, verifies and analyzes the specific impact mechanism between them, and makes suggestions for effective countermeasures in this regard.

*2.2. Child Development Indicators of in Rural Shanxi*

According to the previous literature, the construction of a child development evaluation index system primarily focuses on macro-level considerations, such as educational acquisition, living environment, health, etc. In spite of the fact that the macro-level index system is not appropriate for this study, it offers ideas for establishing a rural child development evaluation index system. Ya-Ju Chang (2015) [18] had also established the Sustainable Child Development Index (SCDI) at present addresses health, education, safety, economic status and environmental aspects described by 25 indicators to provide a more comprehensive evaluation of sustainable development (As the SCDI is a general index for all countries, it may result in an incomplete and biased assessment of the sustainable development status of each country). The UN Task Team Report on the Post-2015 UN Development Agenda indicate that sustainable development involves progress both within and across four integrally connected dimensions: inclusive social development, inclusive economic development, environmental sustainability; and underpinning all of these, the rule of law. Each of these dimensions has specific influences and major impacts on children. At the same time, children are central to the progress and contributions of each of these four dimensions [19].

Therefore, this paper, considering the basic connotations of child development and the current situation of rural children in Shanxi, concludes that the rural child development index system at the first level should consider four components: child health development, educational development, behavioral development, and social development.

At the second level, physical health and mental health are selected as measures of a child's healthy development; academic achievement and educational expectations are selected to reflect a child's educational development; behavioral disorders and conduct disorders are selected to reflect children's behavioral development; and self-concept and social skill are selected to reflect a child's educational development. The second level indicators are measured according to the meaning of the indicator and the specific questions in the questionnaire.

Table 1 is the rural child development evaluation indicator system framework, this study intends to use the number of "fever, cold, asthma, dyspnea, headaches and other diseases in the past 30 days", ranging from 0~64 (the lager the number is, the worse body condition of child is), in the questionnaire as an indicator of the health of rural children; measures academic achievement using the respondents' total scores in mathematics and Chinese; reflects children's educational expectations by asking students about their educational goals; Measure behavioral disorders by asking four questions about self-control and attention; Measure conduct disorders by asking six questions about children fighting, abusing their classmates, intentionally damaging others' property, etc.; Measure Social skills by asking questions about students' interactions with parents, classmates, teachers, etc.

**Table 1.** The Rural Child Development Evaluation Indicator System Framework.

| 1st Level Indictor | 2nd Level Indictor | Measurement |
|---|---|---|
| health development | physical health | The frequency of illnesses occurs in a month? |
| | mental health [1] | The frequency with which you experience sadness, depression, sadness, loneliness, etc. |
| educational development | academic achievement [2] | the total scores in mathematics and Chinese |
| | educational expectations [3] | the educational goals |
| behavioral development | behavioral disorders [4] | self-control and attention |
| | conduct disorders [5] | children fighting, abusing their classmates, intentionally damaging others' property, etc.; |

**Table 1.** *Cont.*

| 1st Level Indictor | 2nd Level Indictor | Measurement |
|---|---|---|
| social development | social skill [6] | Have you discussed your recent circumstances with your mother (or father, teacher, friend)? |
| | self-concept [7] | How do children rate their own problem-solving abilities, their confidence, and their image of themselves? |

[1] Specifically, the questions ask students to provide sentences that describe their feelings over the past two weeks. Examples include I occasionally feel unhappy, I feel unhappy often, and I feel unhappy always. The options consist of three numbers between 1 and 3, with higher numbers indicating more serious negative emotions. [2] Considering the needs of this paper and the availability of data, we use the total score of the subjects in mathematics and Chinese as the measurement of academic performance, resulting in a continuous variable between 17 and 190 points. Higher scores are associated with better academic development. [3] Students are asked about what level of education they anticipate achieving in the questionnaire. The specific options include primary school, junior high school, high school, vocational high school, college, and unclear. [4] Based on the definitions and questionnaires for behavioral disorders in children, this paper evaluates the presence of behavioral disorders in children. The options are 1–6 points, plus a total score ranging from 4 to 24 points. Generally speaking, the higher the score, the less likely the child is to have a behavioral disorder. [5] As a measure of conduct disorder, this article uses 6 questions, such as whether children fight, abuse classmates, damage other people's belongings, etc. The options are 1–5 points, and the total is 6–30 points. The larger the value, the more severe the disorder. [6] This article uses the questions in the questionnaire to examine students' interactions with parents, classmates, and teachers. The specific question is "How often have you spoken to your mother (or father, teacher, friend) about your recent situation in the past 30 days?" The options have 1–6 points, and the total is a continuous variable of 4–24 points. The higher the score, the more frequent the conversation, and the stronger the child's social skills. [7] To measure the self-concept dimension of rural children, the questionnaire includes the questions reflect their self-independent problem-solving ability, self-confidence, self-image, and other 7 aspects. The higher the score, the better the child's self-concept development.

Following the construction of the indicator system framework, this paper used principal component analysis to weight each second-level indicator.

Table 2 below shows the results of the KMO test on each secondary index in order to verify whether it is suitable for principal component analysis:

**Table 2.** The KMO and Bartlett tests.

| KMO Measure of Sampling Adequacy | | 0.828 |
|---|---|---|
| Bartlett's test of sphericity | Approx. Chi-Square | 322.101 |
| | df | 28 |
| | Sig. | 0.000 |

The KMO value of the test result is 0.828 > 0.7 ($p < 0.05$). This shows that each secondary index is suitable for principal component analysis. Then we perform principal component analysis on each secondary index to determine its eigenvalue, variance contribution rate, and cumulative variance contribution rate, as shown in Table 3. Table 3 also shows that the cumulative variance explanation rate of the first 6 principal components is approximately 84%, which can explain the majority of information in this paper.

**Table 3.** Principal components analysis.

| | Eigenvalue | % Total Variance | % Cumulative |
|---|---|---|---|
| Comp1 | 1.933 | 23.830 | 23.830 |
| Comp2 | 1.212 | 15.019 | 38.849 |
| Comp3 | 1.012 | 13.203 | 52.052 |
| Comp4 | 0.956 | 11.322 | 63.374 |
| Comp5 | 0.917 | 10.787 | 74.161 |

**Table 3.** *Cont.*

|  | Eigenvalue | % Total Variance | % Cumulative |
|---|---|---|---|
| Comp6 | 0.774 | 10.122 | 84.283 |
| Comp7 | 0.649 | 9.062 | 93.345 |
| Comp8 | 0.546 | 6.655 | 100.000 |

Table 4 shows the main component factor loading matrix, that is, the coefficients of the 6 secondary indexes.

**Table 4.** The Rotated Component Matrix.

|  | Comp1 | Comp2 | Comp3 | Comp4 | Comp5 | Comp6 |
|---|---|---|---|---|---|---|
| physical health | 0.308 | −0.078 | 0.81 | 0.25 | 0.368 | 0.043 |
| mental health | 0.078 | 0.620 | −0.115 | 0.699 | 0.019 | 0.148 |
| academic achievement | 0.303 | 0.418 | 0.438 | −0.256 | −0.644 | −0.187 |
| educational expectations | 0.602 | 0.353 | −0.224 | −0.173 | −0.070 | 0.297 |
| behavioral disorders | 0.634 | −0.179 | −0.190 | 0.172 | 0.047 | −0.649 |
| conduct disorders | 0.381 | 0.368 | −0.029 | −0.513 | 0.562 | 0.009 |
| social skill | 0.520 | −0.522 | 0.037 | −0.004 | −0.210 | 0.435 |
| self-concept | 0.758 | −0.156 | −0.166 | 0.183 | −0.017 | 0.046 |

By dividing the coefficients of the principal components by the square root of the corresponding eigenvalues of each principal component, the coefficients of the linear combination of each index in each principal component can be derived in Table 5.

**Table 5.** Coefficient correlation of principal components for the indexes.

|  | Comp1 | Comp2 | Comp3 | Comp4 | Comp5 | Comp6 |
|---|---|---|---|---|---|---|
| physical health | 0.2171 | 0.3892 | 0.4429 | −0.2596 | −0.6724 | −0.2146 |
| mental health | 0.4314 | 0.3287 | −0.2265 | −0.1755 | −0.0731 | 0.3408 |
| academic achievement | 0.2207 | −0.0726 | 0.8191 | 0.2536 | 0.3842 | 0.0482 |
| educational expectations | 0.0559 | 0.5773 | −0.1163 | 0.7089 | 0.0198 | 0.1698 |
| behavioral disorders | 0.4543 | −0.1667 | −0.1921 | 0.1744 | 0.0491 | −0.7447 |
| conduct disorders | 0.5432 | −0.1453 | −0.1679 | 0.1856 | −0.0177 | 0.0528 |
| social skill | 0.3727 | −0.4860 | 0.0374 | −0.0041 | −0.2193 | 0.4992 |
| self-concept | 0.2730 | 0.3426 | −0.0293 | −0.5203 | 0.5868 | 0.0103 |

Then we can obtain the child development composite measurement index model as follows.

$$C = (0.2171 * Comp1 + 0.3892 * Comp2 + 0.4429 * Comp3 - 0.2596 * Comp4 - 0.6724 * Comp5$$
$$-0.2146 * Comp6)/0.8409$$

Lastly, normalize the coefficients in the comprehensive index model to obtain the weight of each second-level index, and the first-level indicator is equal to the sum of the weights of the second-level indexes. From Table 6, it can be seen that educational develop-

ment is weighted 19%, healthy development is weighted 47%, behavioral development is weighted 15%, and social development is weighted 18%.

**Table 6.** The Weight in the Rural Child Development Evaluation Indicator System Framework.

|  | **Weight** |  | **Weight** |
|---|---|---|---|
| health development | 0.19 | physical health | 0.04 |
|  |  | mental health | 0.15 |
| educational development | 0.47 | academic achievement | 0.26 |
|  |  | educational expectations | 0.21 |
| behavioral development | 0.15 | behavioral disorders | 0.02 |
|  |  | conduct disorders | 0.13 |
| social development | 0.19 | social skill | 0.05 |
|  |  | self-concept | 0.14 |

## 3. Empirical Studies

### 3.1. The Data

In this article, data are drawn from a special survey conducted on rural child development in Yushe County, Shanxi Province, in 2019. In response to the preliminary investigation and detailed demonstration, the fourth-graders in all rural primary schools in the county were selected as a sample population. Eight primary schools were selected using staged probability proportional sampling (PPS). Some fifth-grade students were engaged in the surveys and interviews in order to gather information concerning children's educational experience, health status, social relations, and teacher evaluation. Overall, the survey collected 860 data on children and 430 data on their actual caregivers.

### 3.2. The Model

Initially, this study used the family structure as an independent variable and a comprehensive development index of rural children in Shanxi China as a dependent variable to develop a benchmark model 1 to study the direct effects of family structure on the development of rural children. Because the development of children in rural China also involves four factors: health, education, socialization, and behavioral development, this study constructs Model 2, Model 3, Model 4, and Model 5 to examine the impact of family structure on children's health, education, behavior, and social development.

The model is constructed as follows:

$$Y_i = \alpha + \sum_{k=1}^{n} \beta_{k1} X_{ki} + \varepsilon_1$$

Here, $Y_i$ represents the development level of student $i$, $X_{ki}$ refer to the $k$-th individual-level variables of student $i$ (including control variables such as family structure, gender, age, etc.). $\beta_{k1}$ represents the $k$-th individual-level regression coefficient; $\alpha$ indicates a fixed intercept; and $\varepsilon_1$ indicates a random disturbance.

Secondly, this paper investigates the mediating effect of parental emotional involvement and behavioral involvement on children's development in Shanxi, rural China. Here, we examine the mediation effects using the bootstrap method proposed by Hayes [20]. The specific model is as follows:

$$Y = i_1 + c_1 D_1 + c_2 D_2 + \ldots c_{k-2} D_{k-2} + c_{k-1} D_{k-1} + \varepsilon_2$$
$$M = i_2 + a_1 D_1 + a_2 D_2 + \ldots a_{k-2} D_{k-2} + a_{k-1} D_{k-1} + \varepsilon_3$$
$$Y = i_3 + c'_1 D_1 + c'_2 D_2 + \ldots c'_{k-2} D_{k-2} + c'_{k-1} D_{k-1} + bM + \varepsilon_4$$

Here, $D_1, D_2, \ldots, D_{k-1}$ represent $k-1$ dummy variables after recoding $k$-level independent variables; $Y$ is the dependent variable; $M$ is the mediator variable; $c_1, c_2, \ldots, c_{k-1}$ represent $k-1$ relative total effects; $c'_1, c'_2, \ldots, c'_{k-1}$ represent $k-1$ relative direct effects.

In this paper, family structure is measured by examining the living arrangements between the students and their parents and is categorized into four groups: (1) complete families, (2) absent father families, (3) absent mother families, and (4) absent parent families. Table 7 indicates that among the families of primary school students surveyed in the sample area, the proportion of complete families is 39.53%, the proportion of absent fathers is 22.67%, the proportion of absent mothers is the lowest, at 7.21%, and the proportion of both parents absent at the same time is 30.58%. In the study, 60.47% of rural primary school children lived in families where both parents were absent or one parent was absent. The rural areas of Shanxi China show a diverse development pattern in terms of family structure and living arrangement, and the proportion of incomplete families is relatively high.

**Table 7.** The family structure.

| Family Structure | Numbers | Sharing |
|---|---|---|
| complete families | 340 | 39.53% |
| absent father families | 195 | 22.67% |
| absent mother families | 62 | 7.21% |
| absent parent families | 263 | 30.58% |

The mediating effect studied in this paper is the mechanism of participation deprivation, which consists of two components: emotional participation and behavioral participation. In the survey, students provide their parents with options from never = 1, rarely = 2, sometimes = 3, often = 4, and always = 5.

The control variable in this study is selected based on characteristics of individual students (or families). It includes the child's gender (male = 1), age, child's self-assessed health, parent-child relationship (frequency of quarrels between you, father and mother, never-always = 1–5), parents' marital quality (father and mother, never-always = 1–5), and peer group quality.

Based on the descriptive statistics presented below (Table 8), the rural family structure in Shanxi shows a diversified pattern, with a relatively high proportion of incomplete families, and incomplete families showed significant differences in their healthy development and social development. Furthermore, compared with complete families, rural children with absent mothers display significantly different comprehensive development than those with absent parents, but no significant differences are detected in families with absent fathers. Furthermore, there is a significant difference in parental behavioral participation between intact families and those with absent fathers, absent mothers, and absent parents.

**Table 8.** The descriptive statistics.

| | Mean | S.D. |
|---|---|---|
| Child Development | 62.45 | 18.46 |
| health development | 83.96 | 14.52 |
| educational development | 71.35 | 17.98 |
| behavioral development | 70.02 | 12.06 |
| social development | 57.25 | 20.11 |
| physical health | 10.33 | 11.25 |
| mental health | 19.22 | 4.12 |
| academic achievement | 137.89 | 32.77 |

**Table 8.** *Cont.*

|  | **Mean** | **S.D.** |
|---|---|---|
| educational expectations | 14.23 | 1.12 |
| behavioral disorders | 19.43 | 5.12 |
| conduct disorders | 28.14 | 3.73 |
| social skill | 11.66 | 4.77 |
| self-concept | 27.25 | 5.78 |

*3.3. Results*

3.3.1. The Effects of Family Structure on the Development of Children in Shanxi Rural Areas

We conducted a regression analysis using the child development index as the dependent variable and the family structure as the independent variable. As the family structure is a four-category independent variable (K = 4), the complete family is used as the reference group, and the type of family structure is dummy coded. Thus, with the entire family as a benchmark, each dummy coding group is used as an independent variable, and the rural child development index, rural children's healthy development index, education development index, behavior development index, and social development index are used as factors in the regression analysis. The result is as follows:

Model 1 indicate that after controlling for individual characteristics such as age, gender, children's self-rated health, peer quality, parent-child relationship, and marital relationship, children growing up in families with absent mothers and in families with both parents absent significantly lag behind children in completed family. This result also partially support hypothesis 1.

The results of Model 2 show that children from non-intact families are more likely to be disadvantaged with respect to healthy development than children from intact families. Specifically, the healthy development for children in absent father families was lower than that of children from intact families; the healthy development for children in absent mother families was also lower than that of children in intact families, and the healthy development for children in families with absent parents was even lower than that of children in complete families. The above results confirm hypothesis 1a. In rural areas, the healthy development of children from incomplete families is significantly lower than that of children from complete families.

Based on the results of Model 3, there was no significant difference between children from incomplete and complete families in their educational development. Model 4 showed that there were no significant differences in the behavioral development of children in families with absent parents, absent fathers, and absent mothers. The results of Model 5 show that the absence of one or both parents will adversely affect the social development of children. In particular, the social development of children in families with absent fathers, absent mothers, and absent parents is 6.23, 5.93, and 8.50 lower than that of children in intact families. The results confirm Hypothesis 1d that the level of social development of rural children from incomplete families is significantly lower than that of children from complete families.

It can therefore be concluded that family structure is indeed closely related to children's development in Shanxi rural areas. The absence of one parent or both parents will have a negative impact on the health and social development of children. A child's overall development can be affected by the absence of a mother as well as the absence of both parents. Due to the differences in roles and functions played by fathers and mothers in the family, there are also significant differences within families without one parent.

### 3.3.2. The Mediating Effect of Parental Emotional Involvement

Using the bootstrap method, this paper considers the entire family as the benchmark level and performs dummy coding on other family types (D1, D2, D3), and then it establishes a regression equation to determine whether parental emotional involvement mediates the relationship between the family structure and the development of children in Shanxi rural areas. Figure 1 illustrates the model result.

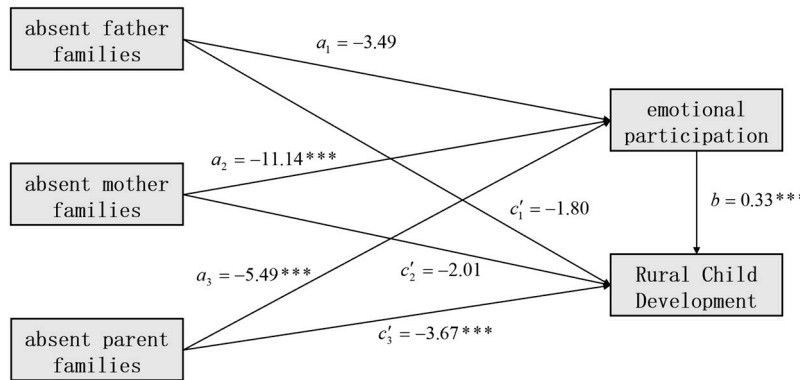

**Figure 1.** The results of mediation model. Note: *, **, and *** denote significance at the 10%, 5%, and 1% levels, respectively.

As a result of the overall mediation analysis, the 95% bootstrap confidence interval for the overall mediation test is [0.27 0.39], excluding 0, suggesting that the three relative mediation effects are not all zero. Therefore, a further relative mediation test analysis is necessary. The relative mediation analysis shows that using the complete family type as a reference, the relative mediation effect for absent father families relative to complete families has a bootstrap confidence interval of 95% [−2.60 0.27], including 0, showing that the relative mediation effect is not significant. Similarly, the 95% bootstrap confidence interval for the relative mediation effect of mother-absent families relative to intact families is [−6.19 −1.49], excluding 0, showing a significant relative mediation effect, and the mediation effects −3.68 ($a_2 = -11.14$, $b = 0.33$, $a_2b = -3.68$), which indicates that the emotional involvement of parents of children from families with absent mothers is 11.14 less than that of children from intact families ($a_2 = -11.14$), so the child development index of the mother-absent family is lower ($b = 0.33$).

In contrast to the direct effect in Table 9 ($c_2 = -5.56$, $p < 0.01$), the relative mediating effect of absent mother families on rural child development was not significant ($c_2' = -2.01$, $p > 0.1$), indicating that parental emotional involvement is the only mediating variable.

**Table 9.** The effects of family structure on the development of children in Shanxi rural areas.

|  | Model 1 | Model 2 | Model 3 | Model 4 | Model 5 |
|---|---|---|---|---|---|
| absent father families | −2.88 | −2.78 * | −0.34 | −1.09 | −6.23 *** |
| absent mother families | −5.56 ** | −4.32 * | −4.82 | −0.45 | −5.91 ** |
| absent parent families | −5.33 *** | −4.89 *** | −0.48 | −0.39 | −8.50 *** |
| Control Variable | Control | Control | Control | Control | Control |
| $R^2$ | 0.28 | 0.16 | 0.09 | 0.11 | 0.19 |
| F-value | 24.74 | 12.58 | 6.94 | 6.67 | 14.33 |

Benchmark group: complete families. Note: * $p < 0.05$, ** $p < 0.01$, *** $p < 0.001$.

Finally, the 95% confidence interval of the relative mediation effect of a family with absent parents versus a complete family with both parents is [−3.23 −0.58], including 0, indicating that there is a significant relative mediation effect. The relative mediating effect

is $-1.81$ ($a_3 = -5.49$, $b = 0.33$, $a_3b = -11.81$), that is, the emotional involvement of parents of children from families with absent parents is 5.49 lower than that of children from families with two parents ($a_3 = -5.49$); consequently, the development index of children growing up in families with absent parents is also lower ($b = 0.33$). The relative direct effect was significant ($c_3' = -3.67$, $p < 0.01$), which indicate that following removal of mediating effects, the development index of children with absent parents was 3.67 lower than that of children in complete families; The effect was significant ($c_3 = -5.33$, $p < 0.01$), and the relative mediation effect of a3b was 33.96% ($-1.81/-5.33$).

In Shanxi rural areas, parental emotional involvement plays a mediating role between family structure and child development, and this effect is significant in mother-absent families and two-parent-absent families, meaning that compared with complete families, mother-absent families and two-parent absent families have lower parental involvement and comprehensive development levels for their children. This partially supports Hypothesis 2a.

### 3.3.3. The Mediating Effect of Parental Behavioral Participation

Similarly, the bootstrap method was also used to examine the mediating role of parental involvement in rural child development. Figure 2 illustrates the model result.

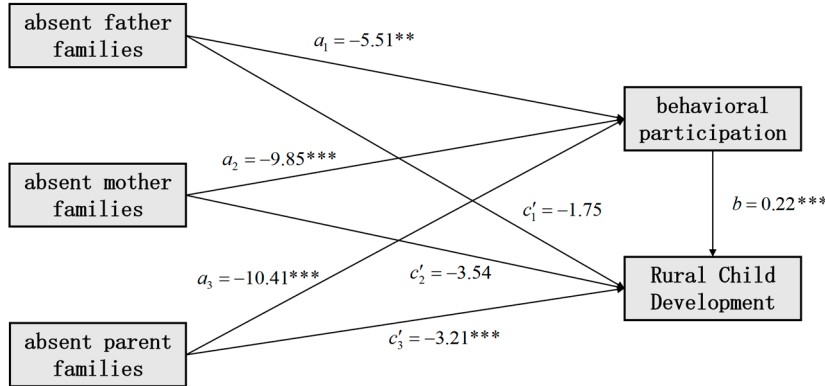

**Figure 2.** The results of mediating effect of parental behavioral participation. Note: *, **, and *** denote significance at the 10%, 5%, and 1% levels, respectively.

The overall mediation analysis shows that the 95% bootstrap confidence interval for the overall mediation test is [0.16 0.27] excluding 0, showing that the 3 relative mediation effects are not all 0. Thus, a further relative mediation test analysis is necessary. Results of the relative mediation analysis indicate that with the complete family type as the reference level, the 95% bootstrap confidence interval of the relative mediation of the father-absent family relative to the complete family is [$-2.38$ $-0.16$], excluding 0, showing that the relative mediation is significant. The 95% bootstrap confidence interval of the relative mediation of the mother-absent family relative to the complete family is [$-4.01$ $-0.53$], excluding 0, showing that the relative mediation is significant, and the 95% bootstrap confidence interval of the relative mediation of the both-parents-absent family relative to the complete family is [$-3.44$ $-1.27$], excluding 0, showing that the relative mediation is significant.

As a result, parental behavioral participation can moderate the relationship between family structure and the development of children in Shanxi rural areas. There is a lower level of parental behavioral participation in fatherless families, mothers' absence families, and families with both parents absent. As a result, the comprehensive developmental level of children growing up in these three types of incomplete families is also lower. Here Hypothesis 2b is confirmed.

*3.4. Robustness Check*

Based on the above analysis, we can conclude that family structure is closely related to the development of children in Shanxi rural areas. An absence of one or both parents will negatively affect the child's health, social development, and overall growth. In the mechanism of family structure affecting child development, parental involvement and behavior clearly have a mediating effect. This paper will conduct a robustness test in order to verify the reliability of the research results as follows.

We first reclassify the four independent variables of complete family, absent father family, absent mother family, and absent parent family into two categories, complete family and incomplete family, for regression analysis and intermediary mechanism testing.

Then, 50% of the total sample will be randomly selected to examine the robustness of parental emotional and behavioral involvement in the development of children in Shanxi rural areas as a mediating mechanism.

Ultimately, we found that the results of both methods reveal that family structure affects children's development in Shanxi rural areas, and that parental emotional and behavioral involvement has a mediated effect between the family structure and the development of rural children in Shanxi China. The conclusion maintains a high degree of reliability.

## 4. Discussion

*4.1. Key Findings*

The rural areas of Shanxi, where population outflows are frequent, have many families with absent parents, which has become a social phenomenon that is difficult to ignore. It may be possible, therefore, to resolve the problem of unbalanced and insufficient development of rural children in Shanxi through discussing the impact of family structure on the growth and development of rural children in Shanxi China as well as promoting their level and capability of development from the perspective of supporting family development.

In terms of the health, education, behavior, and social development of rural children, children from incomplete families are significantly different from children from complete families. In the current context of the diversification of rural family structures in Shanxi, it can be demonstrated that the family structure is closely related to the development of rural children in Shanxi, regardless of whether it is a comprehensive study or a separate investigation. Similar with Collyer's views [21,22], the emotional and behavioral involvement of parents plays a significant mediating role between family structure and rural child development in Shanxi. Specifically, from an emotional perspective, children who grow up with both parents can receive positive emotional input from both parents promptly, which can promote the healthy development of children in all aspects. From a behavioral perspective, in a complete family, both parents have enough time and energy to participate in their children's academic activities and to establish a favorable tutoring environment by tutoring their children's homework and inquiring about their daily living conditions. These behaviors promote children's healthy growth. So future rural policies should also provide more support to improve farmers' incomes and promote the employment and entrepreneurship of returning migrant workers.

To promote rural child development in Shanxi, it is essential to focus on families, schools, and rural communities and to integrate external support resources such as schools and rural communities to form a comprehensive management system for rural child development that includes family development as well as school development. It should be noted that due to the prevalence of the traditional family division of labor in Shanxi China, women continue to play a vital role in caring for the family and ensuring the health and development of their children. So in rural areas, it is therefore essential to provide rural female laborers with opportunities for employment in close proximity and to increase their participation in the growth and development of their children, as this will contribute to the continuous improvement of the comprehensive level of development of those children.

*4.2. Limitations and Outlook for Future Research*

Based on previous studies, this paper establishes a rural child development index system and examines the effects of family structure on rural child development in Northwest China. In addition, the paper analyses and verifies the mechanisms by which the family structure impacts the development of rural children. It should be noted, however, that this study has the following two limitations.

4.2.1. Universality of the Indicator System

It may not be as significant a problem in the central and eastern regions of China than in the northwest, where frequent population movements are responsible for the problem of rural children's development. For example, in western China, the proportion of left-behind children living with their grandparents exceeds 50%, which is much higher than in central and eastern China. According to the "e-Growth Rural Children's Internet Use Status Report" (Reports of Guangdong Foundation for Poverty Alleviation in China, 2021), there is also a "digital divide" between rural children in the west and the east. The differences are not only evident in the equipment, time, and content of Internet access, but also in Internet skills, literacy, and family education methods. Children living in rural western areas use the internet at an earlier age, with less supervision, and high risk, as opposed to children living in rural eastern areas. Therefore, the environment of children in rural areas varies greatly within China.

Thus, it is necessary to explore the universality of the rural child development indicator system for further purposes, additional topics, indicators and data regarding rural child development.

4.2.2. Limited Data Availability

The low data availability leads to limited consideration of indicators within specific topics. For example, we define the dependent variable "children's mental health" in accordance with the questionnaire responses. However, this requires children to possess advanced cognitive and emotional abilities. Due to this, a rural fourth-grade sample was selected for this study in order to ensure the validity of the empirical results and the reliability of the results. There is no doubt that this has some limitations, which necessitate the expansion of the research sample in the future.

Additionally, considering that the data from the rural children's questionnaire used in this study are not tracked survey data, future research should examine how changes in family structure continue to affect the development of rural children.

**Author Contributions:** Conceptualization, Z.L.; methodology, Z.L.; formal analysis, H.L.; investigation, H.L.; resources, H.L.; writing—original draft preparation, H.L.; writing—review and editing, Z.L.; supervision, Z.L. All authors have read and agreed to the published version of the manuscript.

**Funding:** This research received no external funding.

**Institutional Review Board Statement:** Not applicable.

**Informed Consent Statement:** Informed consent was obtained from all subjects involved in the study.

**Data Availability Statement:** The data presented in this study are available on request from the corresponding author. The data are not publicly available due to privacy.

**Conflicts of Interest:** The authors declare no conflict of interest.

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
