# Peer review of "Family Environment and Rural Child Development in Shanxi, China"

_sustainability, doi:10.3390/su142013183_

Round 1

Reviewer 1 Report

Before resubmission (anywhere), the authors must address at least the following key comments:

i) I understand that the text is intended for the Sustainable Education and Approaches section, but it is not clear to me how it is consistent with the mission of Sustainability journal. The text does not address sustainability or any aspect of it (indeed, the word sustainability is completely absent from the text). 

ii) The text lacks discussion and does not respect the expected IMRAD structure. A summary is no substitute for discussion. The text lacks clearly defined limits of the study.

iii) The text contains a large number of typos (missing spaces, etc.). This needs to be corrected.

Author Response

  1. I understand that the text is intended for the Sustainable Education and Approaches section, but it is not clear to me how it is consistent with the mission of Sustainability journal. The text does not address sustainability or any aspect of it (indeed, the word sustainability is completely absent from the text). 

Actually, the Child development index in our paper can reflects the Child Well-Being and Sustainable Development. So our index also can be called Child sustainable development index. There are many other similar researches, for example, Ya-Ju Chang(2015), also introduce a similar index called Sustainable Child Development Index.   

Chang Y J, Lehmann A, Winter L, et al. The sustainable child development index (SCDI) for countries[J]. Sustainability, 2018, 10(5): 1563.

  1. ii) The text lacks discussion and does not respect the expected IMRAD structure. A summary is no substitute for discussion. The text lacks clearly defined limits of the study.

We add some content to describe the theoretical method in line 114 to line 150.

iii) The text contains a large number of typos (missing spaces, etc.). This needs to be corrected.

We proofread the paper again and change some typos.

Reviewer 2 Report

This manuscript is a good and well-argumented read. However, I have a few minor queries to respond to: 

1. Abstract is vague. I suggest the authors re-write by adding the method and critical results to clarify it for the general readers.

2. The authors did not mention how they collected the data and interviewed the participants. I suggest they should explain the process of data collection and the interviews.

3. Authors presented the tables and discussed data under heading 5 (line 218). I believe it should come early to set the stage for data analysis.

4. How did the authors design the survey? Did they construct a new tool or use an existing one? How many questions did they include in the survey?

5. The authors should add some discussion to rationalize their results. I could not see any cross-validation of the results. I suggest the authors should refer to some studies to support their new findings.

6. I suggest the authors use a conclusion instead of a summary. The study lacks if it has any limitations. It would add value to the study if the authors could also add future directions for the researchers.

7. I also suggest that the authors go for a language check as it requires minor adjustments. For example, in Table 1, line 172, "2ed Level Indictor". I believe it is an "indicator" and also "2nd".

Author Response

  1. Abstract is vague. I suggest the authors re-write by adding the method and critical results to clarify it for the general readers.

We added some methods and results in the abstract.

  1. The authors did not mention how they collected the data and interviewed the participants. I suggest they should explain the process of data collection and the interviews.

We add some introduction about our questionnaire in the footnote in page 5.(table1)

  1. Authors presented the tables and discussed data under heading 5 (line 218). I believe it should come early to set the stage for data analysis.

 This part is mainly introduce the Child Development indicators. It’s important in our paper. So I split it from Heading 6.

  1. How did the authors design the survey? Did they construct a new tool or use an existing one? How many questions did they include in the survey?

We add some introduction about our questionnaire in the footnote in page 5.(table1)

  1. The authors should add some discussion to rationalize their results. I could not see any cross-validation of the results. I suggest the authors should refer to some studies to support their new findings.

We add some references in the last paragraph.

  1. I suggest the authors use a conclusion instead of a summary. The study lacks if it has any limitations. It would add value to the study if the authors could also add future directions for the researchers.

We add some future directions in the last paragraph.

  1. I also suggest that the authors go for a language check as it requires minor adjustments. For example, in Table 1, line 172, "2ed Level Indictor". I believe it is an "indicator" and also "2nd".

Modified some typos.

Round 2

Reviewer 1 Report

Thank you for changes. Unfortunately, I do not consider them sufficient to publish the text. I will use the previous answers below. New notes are in bold.:

  1. I understand that the text is intended for the Sustainable Education and Approaches section, but it is not clear to me how it is consistent with the mission of Sustainability journal. The text does not address sustainability or any aspect of it (indeed, the word sustainability is completely absent from the text). 

Actually, the Child development index in our paper can reflects the Child Well-Being and Sustainable Development. So our index also can be called Child sustainable development index. There are many other similar researches, for example, Ya-Ju Chang(2015), also introduce a similar index called Sustainable Child Development Index.   

Chang Y J, Lehmann A, Winter L, et al. The sustainable child development index (SCDI) for countries[J]. Sustainability, 2018, 10(5): 1563.

Thank you for the explanation, please explain sufficiently directly in the text of the article. 

  1. ii) The text lacks discussion and does not respect the expected IMRAD structure. A summary is no substitute for discussion. The text lacks clearly defined limits of the study.

We add some content to describe the theoretical method in line 114 to line 150.

Discussion and limits are still lacking. Study the IMRAD structure and complete the discussion chapter. Without it, the article cannot be published. (E.g.: https://writingcenter.gmu.edu/writing-resources/imrad/writing-an-imrad-report)

iii) The text contains a large number of typos (missing spaces, etc.). This needs to be corrected.

We proofread the paper again and change some typos.

Thank you for that.

Author Response

Dear Editors and Reviewers,

Thank you for giving us the opportunity to submit a revised draft of our manuscript. We appreciate the reviewers for their constructive comments and suggestions on my paper.

Please check the attachment for the full paper.

Thanks a lot.

Round 3

Reviewer 1 Report

thank you for sufficiently incorporating my comments